evolution/taxonomy and systematics/palaeontology

terrestrialization, fossil arthropods, air-breathing, synchrotron radiation, X-ray fluorescence imaging, X-ray spectroscopy

**Author for correspondence:**
Pierre Gueriau
e-mail: pierre.gueriau@hotmail.fr

# A new Devonian euthycarcinoid reveals the use of different respiratory strategies during the marine-to-terrestrial transition in the myriapod lineage

Pierre Gueriau[1,2], James C. Lamsdell[3], Roy A. Wogelius[4],
Phillip L. Manning[4,5], Victoria M. Egerton[4,5],
Uwe Bergmann[6], Loïc Bertrand[2,7] and Julien Denayer[8]

[1]Institute of Earth Sciences, University of Lausanne, Géopolis, CH-1015 Lausanne, Switzerland
[2]Université Paris-Saclay, CNRS, ministère de la Culture, UVSQ, MNHN, Institut photonique d'analyse non-destructive européen des matériaux anciens, 91192, Saint-Aubin, France
[3]Department of Geology and Geography, West Virginia University, 98 Beechurst Avenue, Morgantown, WV 26505, USA
[4]University of Manchester, Interdisciplinary Centre for Ancient Life, Department of Earth and Environmental Sciences, University of Manchester, Manchester M13 9PL, UK
[5]The Children's Museum of Indianapolis, 3000 N Meridian St, Indianapolis, IN 46208, USA
[6]Stanford PULSE Institute, SLAC National Accelerator Laboratory, Menlo Park, CA 94025, USA
[7]Université Paris-Saclay, ENS Paris-Saclay, CNRS, Photophysique et Photochimie Supramoléculaires et Macromoléculaires, 91190 Gif-sur-Yvette, Saint-Aubin, France
[8]Evolution and Diversity Dynamics Lab, Geology Research Unit, University of Liège, Allée du Six-Août, B18, Sart Tilman, B4000 Liège, Belgium

PG, 0000-0002-7529-3456; JCL, 0000-0002-1045-9574;
RAW, 0000-0002-5781-2152; PLM, 0000-0002-7161-6246;
VME, 0000-0002-0739-6533; UB, 0000-0001-5639-166X;
LB, 0000-0001-6622-9113; JD, 0000-0002-4339-7760

Myriapods were, together with arachnids, the earliest animals to occupy terrestrial ecosystems, by at least the Silurian. The origin of myriapods and their land colonization have long remained puzzling until euthycarcinoids, an extinct group of aquatic arthropods considered amphibious, were shown to be stem-group myriapods, extending the lineage to the Cambrian and evidencing a marine-to-terrestrial transition. Although possible respiratory structures comparable to the air-breathing tracheal system of myriapods are visible in several euthycarcinoids, little is known about the mechanism by which they respired. Here, we describe a new euthycarcinoid from Upper Devonian alluvio-lagoonal deposits of Belgium. Synchrotron-based elemental

X-ray analyses were used to extract all available information from the only known specimen. Sulfur X-ray fluorescence (XRF) mapping and spectroscopy unveil sulfate evaporation stains, spread over the entire slab, suggestive of a very shallow-water to the terrestrial environment prior to burial consistent with an amphibious lifestyle. Trace metal XRF mapping reveals a pair of ventral spherical cavities or chambers on the second post-abdominal segment that do not compare to any known feature in aquatic arthropods, but might well play a part in air-breathing. Our data provide additional support for amphibious lifestyle in euthycarcinoids and show that different respiratory strategies were used during the marine-to-terrestrial transition in the myriapod lineage.

## 1. Introduction

The establishment of complex terrestrial ecosystems (terrestrialization) is a critical event in the history of life, which imposed multiple constraints upon the water-to-land transition of aquatic organisms, such as osmoregulation (water balance to avoid dehydration), respiration and reproduction in air, as well as locomotion without the help of buoyancy, and exposure to ultraviolet radiation (e.g. [1] and references therein). It is therefore not surprising that arthropods were the first animals to take the step, with their exoskeleton comprising a waxy layer that controls water loss and facilitates osmoregulation, and their jointed appendages supporting surface locomotion ([1] and references therein). However, air-breathing is more challenging, as the gill-modified appendages of aquatic arthropods are not rigid and are highly vulnerable to dehydration. In general, respiratory organs invaginate in terrestrial organisms, while they evaginate in aquatic ones (e.g. [2]). Pulmonate arachnids (e.g. scorpions, whip scorpions and spiders) and (semi-)terrestrial crabs have internalized and/or adapted their gills, respectively, into book lungs found inside open ventral abdominal air-filled cavities ([1,3] and references therein), and through gill stiffening and gill chamber enlargement and/or smooth to highly convoluted lining (e.g. [4]). Terrestrial isopods (including woodlice) use their gills, formed by part of their abdominal legs (endopods), for respiration; some groups also modify exopods into invaginations to form lung-like structures ([1,3] and references therein). Myriapods (e.g. centipedes and millipedes), apulmonate arachnids (e.g. harvestmen, mites, ticks and pseudoscorpions) and terrestrial hexapods (bristletails, silverfish and insects) except for some springtails (which exchange gases by diffusion across the cuticle, a capability linked to their small size), independently evolved an entirely new respiratory system, the tracheal system, which consists of small pores (the spiracles) opening on an extensive and elaborate system of air-tubes (the tracheae).

These respiratory organs are already present in the earliest known terrestrial animals, the myriapods and arachnids, which colonized terrestrial ecosystems at least by the Silurian [5,6]. One of the oldest known myriapods, the millipede (Diplopoda) *Pneumodesmus newmani* (Wilson & Anderson, 2004) from the Early Devonian (*ca* 414 Ma; [7,8]) of Scotland, very likely possessed a tracheal system as suggested by the presence of slit-like spiracle openings [6]. Book lungs indistinguishable from those in modern Arachnida have been found in trigonotarbids (an extinct group of spider-like arachnids) from the Early Devonian (*ca* 410 Ma) Rhynie chert hot spring complex in Scotland [9]. Book lungs in the scorpion *Pulmonoscorpius kirktonensi* (Jeram, 1994) from the Lower Carboniferous of Scotland show similarities both with those in modern scorpions and *Limulus* book gills suggesting that arachnid book lungs are derived from internalized book gills [10]. Gene expression data and embryology of modern scorpions, however, showed that part of the book lung (the operculum) is derived from the walking legs of abdominal appendages (telopodites), rather than from epipods (the outermost ramus of appendages) as for book gills [11]. The fossil record further yields insights into the origin and marine history of arachnids (Chelicerata). It can be traced as far back as the early Cambrian [12–15], and even documents marine-to-terrestrial adaptations in scorpions [16]. By contrast, the water-to-land transition in the myriapod lineage is mostly a mystery as (i) the oldest known fossils are terrestrial and show evidence for a tracheal system [6] making it impossible to know whether the latter arose during terrestrialization or whether it replaced an earlier respiratory organ [3], and (ii) no aquatic or terrestrial stem group of Myriapoda had been recognized until very recently [17]. The identification of the extinct euthycarcinoids as aquatic stem-group myriapods extends the myriapod lineage to the Cambrian [17], as suggested by fossil-calibrated molecular phylogenies [18,19], and provides the first insights into the water-to-land transition in Myriapoda [17].

Euthycarcinoids form an extinct group of rare (only 18 taxa formally described), and often overlooked, arthropods extending from the Cambrian [20,21] to the Middle Triassic [22–25]. Cambrian

taxa were marine [20,21], younger forms inhabited brackish to freshwater environments [26]. They have been alternatively interpreted as related to crustaceans, xiphosuran chelicerates, hexapod-like arthropods or transitional between myriapods and hexapods (see [26] and references therein). A position as stem-group myriapods had recently emerged as the most plausible based on Bayesian phylogenetic analysis [27,28] and is now supported by new, high-resolution details of head structures from the Early Devonian euthycarcinoid *Heterocrania rhyniensis* (Hirst & Maulik, 1926) from the Rhynie and Windyfield cherts [17]. Euthycarcinoids are considered amphibious and were probably one of the earliest animals to venture onto land, providing unique insights into terrestrial adaptations in the myriapod lineage. Coastal trackways from the Cambrian [21,29–31] suggest that they were indeed capable of at least brief terrestrial excursions [1,26], yet the mechanism by which euthycarcinoids aerially respired remains poorly understood. Several euthycarcinoids display a pair of ventral pores per pre-abdominal somite, located anteriorly to internal tube-like structures [25,32,33] resembling the sternal apodemes of progoneate myriapods. Edgecombe & Morgan [25], however, interpreted such 'sternal pores' in *Synaustrus brookvalensis* (Riek, 1964) as vesicles having a role in minimizing water loss, rather than as spiracles of a tracheal system. Here, we report a new Upper Famennian (Upper Devonian) euthycarcinoid from the very shallow-water alluvio-lagoonal deposits of Belgium that possesses a pair of ventral spherical cavities or chambers on the post-abdomen forming, most likely, a different air-breathing system.

# 2. Material and methods

## 2.1. Origin of the fossil material

The euthycarcinoid fossil was collected in the 1970s by Prof. Édouard Poty (Liège University) from the Upper Famennian (Evieux Formation) strata at Pont de Bonne, Belgium (figure 1), in an outcrop famous for its fossil plants [36] and eurypterids [37,38] but nowadays partly inaccessible and walled (see [37] for a complete description of the section, and electronic supplementary material, geological background). At Pont de Bonne, the Evieux Formation is interpreted as showing both tidal flat and lagoonal settings [39], locally developed as a slightly dolomitic shale with *Racophyton* remains. It is commonly acknowledged that this fern grew in brackish water marshes [40]. The euthycarcinoid fossil comes from such a *Racophyton* dolomitic shale. Contrary to Fraipont's material [37], the specimen is fully articulated indicating very little post-mortem (or post-moulting) transport and presents an uncommon limonitic preservation, similar to that of plant remains coming from the same bed. No counterpart was found, and the specimen is preserved in ventral view. Only the margins and dorsal carinae (preserved as negative reliefs, supporting the interpretation of the fossil as preserved in ventral view) of the pre-abdomen cuticle are visible, indicating that the specimen displays the internal view of the tergites. The post-abdomen and telson cuticle, however, preserves complete surfaces and therefore displays the external view of the sternites.

## 2.2. Synchrotron analyses

The chemical composition of the fossil was investigated using synchrotron rapid scanning X-ray fluorescence (SRS-XRF) mapping at the wiggler beamline 6–2 of the Stanford Synchrotron Radiation Lightsource (SSRL) (figure 2). The diameter of the incoming X-ray beam was reduced to 50 μm using a pinhole (see [41,42] for detailed information about the beamline optics). The fossil was mounted on an x-y scanner stage allowing large movements relative to the X-ray beam with micrometric accuracy. Fluoresced photons were collected using a single element XRF Vortex silicon drift detector placed in the horizontal plane. Integrated intensities in preselected spectral regions corresponding to the emission lines of elements of interest (electronic supplementary material, figure S1) were recorded on the fly in the horizontal direction at a pixel rate corresponding to a scan distance of 50 μm per approximately 4 ms. Full XRF spectra were additionally collected in several points of interest with a 30 s count time (electronic supplementary material, figure S1). Distributions of Ca, Mn, Fe, Ni, Cu, Zn, Ga, As, Br ($K\alpha_1$) and Pb ($L\beta_1$) XRF lines were mapped using an incident beam energy of 13.5 keV. Note that fitting of the full XRF spectra using the PyMCA data-analysis software [43] indicates that (i) although largely dominated by As, the As distribution also includes signal arising from Pb ($L\alpha_1$ line, which falls in the same energy domain as the As $K\alpha_1$ line), (ii) the Pb distribution ($L\beta_1$ line) is actually dominated by pile-up signal from iron, and (iii) no Br was detected (electronic supplementary material, figure S1). Distributions of Al, Si, P, S and Cl $K\alpha_1$ XRF lines were mapped using an incident

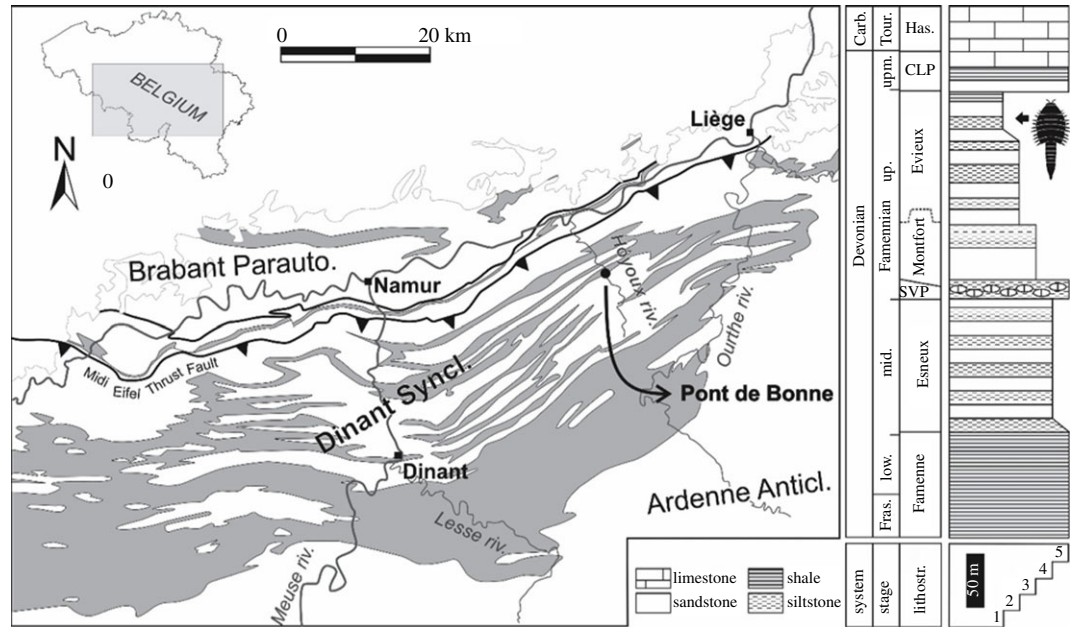

**Figure 1.** Geological and stratigraphical settings of the Pont de Bonne locality, Liège Province, Belgium. Geological map after [34], grey areas represent the Famennian outcrop zones. Famennian lithostratigraphic succession of the Hoyoux valley after [35] with localization of the fossiliferous horizon. Legend: 1, alluvial and alluvio-lagoonal facies; 2, back barrier to lagoonal facies; 3, tidal flat and barrier complex facies; 4, proximal subtidal facies; 5, open-marine subtidal facies. Abbreviations: Anticl., Anticline; Carb., Carboniferous; CLP, Comblain-au-Pont Formation; Fras., Frasnian; Has., Hastière Formation; Lithostr., Lithostratigraphic units; low., lower; mid., middle; Parauto., Parautochton; SVP, Souverain-Pré Formation; Syncl., Syncline; Tour., Tournaisian; up., upper; upm., upmost.

beam energy of 3.15 keV while the fossil was enclosed within a custom-built helium-purged chamber to reduce absorption and scattering of the incident and fluoresced low-energy X-rays by air. All elemental distributions presented herein (figure 2, electronic supplementary material, figures S1, S2) are displayed using a linear grey scale going from white (low abundance) to black (high abundance), normalized between the 10th and 90th percentiles of each distribution (except for the S map, slightly saturated to better reveal S associated with the fossil).

X-ray absorption near edge structure (XANES) spectroscopy at the S K-edge was performed to determine sulfur speciation (figure 2*h*). S XANES spectra were collected in fluorescence mode in the 2450–2550 eV range with energy steps of 1 eV between 2450 and 2460 eV, 0.25 eV between 2460 and 2490 eV, and 1 eV between 2490 and 2550 eV. The count time was set to 0.5 s per energy step. $ZnSO_4$ (prepared as a pellet) was used for energy calibration by setting the position of the main resonance (i.e. sulfate peak) at 2481.75 eV.

# 3. Results

## 3.1. Systematic palaeontology

Arthropoda von Siebold, 1848
Mandibulata Snodgrass, 1938
Euthycarcinoidea Gall & Grauvogel, 1964
Euthycarciniformes Starobogatov, 1988
*Ericixerxes potii* gen. et sp. nov.

### 3.1.1. Etymology

The genus name is from '*ericius*' (Latin for hedgehog), referring to the sharp epimera of the pre-abdomen tergites (gender masculine). The species name honours Prof. Édouard Poty who collected the fossil.

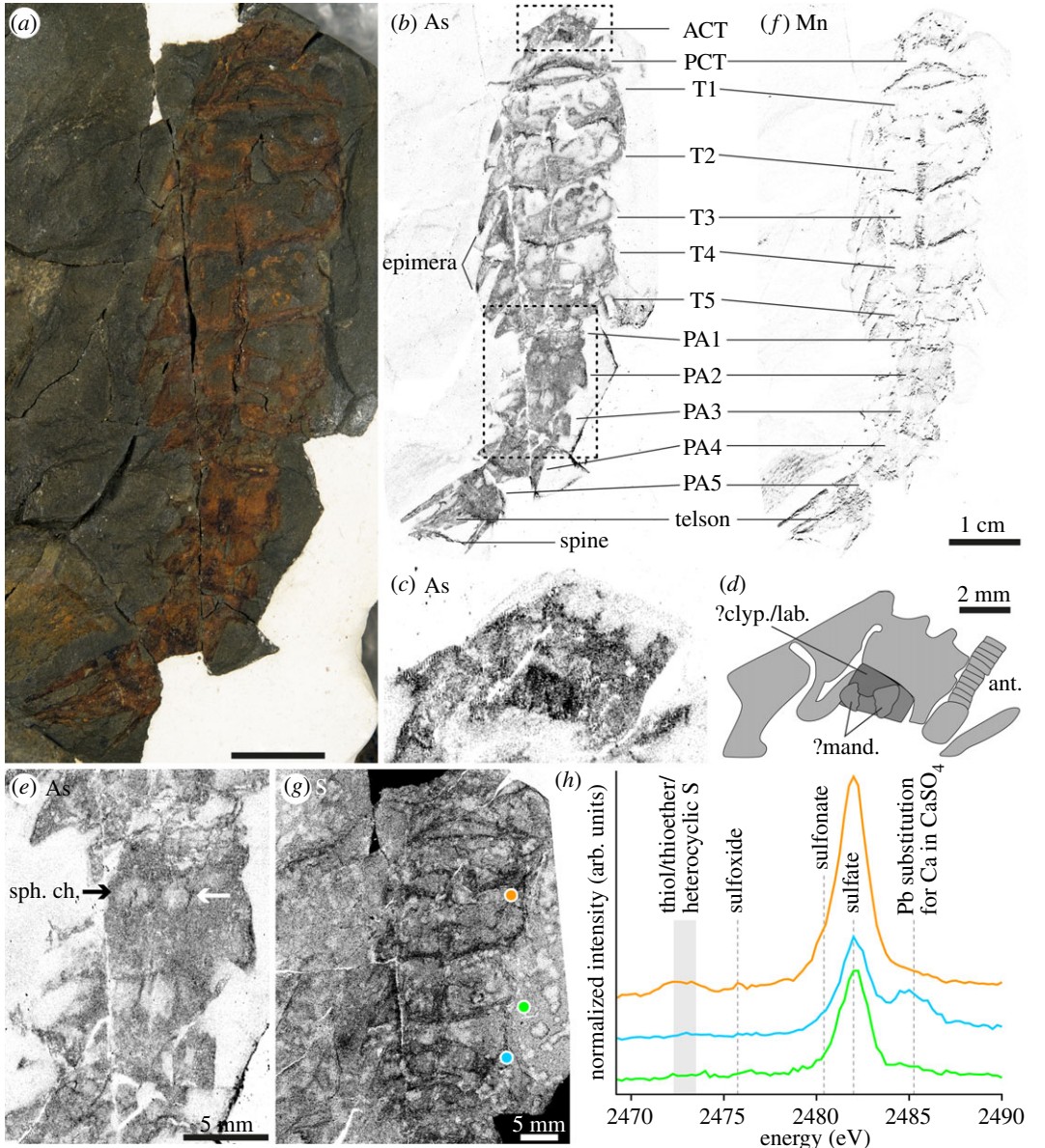

**Figure 2.** *Ericixerxes potii* gen. et sp. nov. (*a*) optical photograph. (*b*–*g*) SRS-XRF elemental distributions of As (*b*–*e*), Mn (*f*) and S (only cephalic region and pre-abdomen; (*g*)) from integrated intensities in their Kα emission energy domains. (*c,d*) Close-up and interpretative line drawing from the top dotted box area in (*b*). (*e*) Close-up from the bottom dotted box area in (*b*). (*h*) XANES spectroscopy at the S K-edge from the areas identified with coloured circles in (*g*), corresponding to the fossil cuticle (orange and blue), and one of the evaporation bubbles (green). Abbreviations: ACT, anterior cephalic tergite; ant., antenna; ?clyp./lab., possible clypeus and/or labrum; ?mand., possible mandibles; PA1–5, post-abdominal somites 1–5; PCT, posterior cephalic tergite; T1–5, pre-abdominal tergites 1–5; sph. ch., spherical chambers. Mapping parameters of the entire maps: scanning step: 50 μm; 901 × 2001 pixels for As and Mn, 801 × 2001 pixels for S; colour scale goes from white (low abundance) to black (high abundance).

### 3.1.2. Stratigraphic age and distribution

Upper Famennian (Upper Devonian) Evieux Formation, Pont de Bonne, Modave municipality, Liège Province, Belgium (50°27′06″ N, 5°17′05″ E).

### 3.1.3. Material

Holotype and only known specimen ULg.PA.2016.07.01 (figure 2*a*), complete individual in ventral view, part only.

### 3.1.4. Diagnosis

Euthycarciniform arthropod with pre-abdomen tergites bearing sharp epimera and a median carina; fifth post-abdominal somite with long and very sharp epimera; second post-abdominal somite with a pair of large, medial circular cavities.

## 3.2. Description

The specimen is 75 mm long from the top of the cephalic region to the end of the post-abdomen (figure 2*a*); maximum width of the pre-abdominal tergites, 19 mm. Semicircular anterior cephalic tergite (ACT) and posterior cephalic tergite (PCT), PCT 1.5 times larger than ACT. Semicircular pre-abdominal tergites (T1–T5) slightly convex posteriorly, with a dorsal median longitudinal carina extending from the middle of the tergite to its posterior margin; at least T2 carina ends in a small spine; T1–T4 of similar dimensions, T5 smaller; T1–T4 bear large and sharp lateral flattened protrusions (epimera), as long as the tergites; T5 with shorter epimera; T1 covering half the PCT; T2–T4 largely covering (*ca* one third) the preceding tergite; T5 covering both T4 and the first post-abdominal somite. Post-abdomen one-half narrower than the pre-abdomen; subrectangular post-abdominal somites (PA1–PA5); PA1–PA4 bear short, blunt epimera; PA5 tapers into long (twice the length of PA5), extremely sharp epimera. Pyriform telson ending in a sharp spine. Cephalic appendages very poorly preserved; abdominal appendages not preserved.

## 3.3. Preservation and palaeoenvironment

The specimen preserves cuticular remains as orangish iron oxides, in a greenish dolomitic shale deposited in brackish water marshes (see §2.1 and supplementary material, geological background). Replacement of fossil tissues by iron oxides often results from the oxidation of pyrite (e.g. [44–46]), which forms in large quantities in such organically rich depositional environments, especially considering that sulfate was present (see below). Besides Fe, major-to-trace elemental mapping using SRS-XRF (figure 2, electronic supplementary material, figure S1, S2) shows that the fossil is also enriched in Cu, As (figure 2*b,c,e*) and to a lesser extent Ca. The fact that the distribution of these elements strictly matches the cuticular remains of the specimen is evidence that it has been controlled (post-mortem) by and reflects original anatomical cuticular structures of the organism. Mn (figure 2*f*), Ni and to a lesser extent Cu (electronic supplementary material, figure S2) delineate the outer and posterior margins of the tergites and sternites, as well as the median carina and telson, allowing a better distinction of the different trunk segments. This indicates thickening of the cuticle margin, very common in arthropods (e.g. in eurypterids [47]). Zn signal is similar in the fossil and in the sedimentary matrix (electronic supplementary material, figure S2), which mainly consists of Al, Si and Fe.

In addition to transition metals, the fossil contains S (figure 2*g*, electronic supplementary material, figure S2). XANES spectroscopy at the S K-edge indicates that S is mostly present as sulfates (figure 2*h*). Traces of organic sulfur, namely cysteine thiol/thioether/heterocyclic S (2472.2–2473.5 eV) and methionine sulfoxide (2475.7 eV), are shown by XANES to be present in several places of the fossil cuticle, most probably representing remnants and/or breakdown products of the chitin-protein complex (figure 2*h*, orange spectrum). Strikingly, the S map reveals bubble-like subspherical to subcylindrical patches enriched in sulfates at their periphery and depleted at their centre, absolutely invisible with regular light (figure 2*g*) and closely resembling evaporation stains ('coffee ring' effect). These stains are present on the entire surface of the sample (both sediment and specimen) but not enriched within fractures (i.e. not secondary infilling), suggesting very shallow-water to subaerial conditions prior to burial.

## 3.4. Additional anatomical details revealed by trace metal distributions

The distribution of As (K$\alpha_1$ emission line: 10.54 keV; probably also including some signal arising from the Pb L$\alpha_1$ line: 10.55 keV, see above) is particularly interesting as its information depth on the order of a couple of hundred micrometres allows unveiling additional anatomical details buried under surface layers (figure 2*b–e*). An elongated and annulated structure originating in the ACT and directed forward very likely represents the left antenna (figure 2*c,d*). Anteriorly to the PCT, a trapezoidal to triangular anteriorly oriented structure, richer in As than the rest of the fossil, stands out as a thicker, or more sclerotized, plate, covering posteriorly two triangular, laterally rounded, processes (figure 2*c,*

*d*). Racheboeuf *et al.* [48], in their redescription of *Schramixerxes gerem* (Schram & Rolfe, 1982) tentatively interpreted a plate with a similar morphology and a similar location on the ventral side of the head as a labrum covering the mouth. This plate also closely resembles the clypeus and/or labrum covering the mandibles in chilopod myriapods (e.g. fig. 4 in [49]), the two triangular laterally rounded processes presumably being the mandibles.

Surprisingly, the As map also uncovers a pair of spherical structures (*ca* 2 mm large) selectively depleted in As (paler areas) and laying ventrally on PA2 (figure 2*b,e*), which were not or hardly visible on the specimen. The clear spherical and symmetrical shape of these structures, not seen elsewhere in the specimen or the sedimentary matrix, make them very unlikely to be the result of diagenetic processes. As their concentration in As is closer to that of the sedimentary matrix than the surrounding cuticle area, it indicates that the cuticle is locally thinner, and that they most likely represent cavities or chambers inside the post-abdomen. There could be a second pair on PA3, but these ones look very different, as they are well visible on the fossil and therefore rather represent lacking parts of the cuticle. They are also larger, closer to each other, and more square, rather similar to the square spaces created by the intersection of the medial carina of T4 and the anterior margin of T5. The pair of spherical structures on PA2 also appears on the Cu map (electronic supplementary material, figure S2).

# 4. Discussion

*Ericixerxes potii* gen. et sp. nov. is only the second Devonian euthycarcinoid known with *H. rhyniensis* from the Early Devonian Rhynie and Windyfield cherts. *E. potii* gen. et sp. nov. and *H. rhyniensis* share a very similar morphology, but *E. potii* gen. et sp. nov. clearly differs by its pre-abdomen bearing large and sharp epimera and a median carina, and its post-abdominal somite 5 with extremely sharp epimera. The peculiarity of *E. potii* gen. et sp. nov. is the pair of quite large, ventral spherical cavities or chambers on the second post-abdominal segment revealed by the distribution of As (figure 2*b,e*). Ventral abdominal cavities or chambers are not known in extant aquatic arthropods.

No pore or opening structure is visible in *E. potii* gen. et sp. nov., whereas some other euthycarcinoids exhibit ventral pores located anteriorly to internal tube-like structures [25,32,33] resembling the sternal apodemes that form the tracheal respiratory system within progoneate dignathan myriapods (these pores have alternatively been interpreted as vesicles minimizing water loss [25]). The tracheal system morphologically varies within Myriapoda. Tracheae differ substantially in their positioning, gross morphology and fine structure (e.g. [50]). Scutigeromorph centipedes have unpaired dorsal spiracles opening into a chamber (the atrium) that gives rise to two large bunches of tracheae bathed in haemolymph [51]. Pleurostigmophoran centipedes exhibit a variable number of pleural spiracles, suggesting that tracheal systems evolved more than once within the Chilopoda [50]. Symphylans have a single pair of spiracles on the head. In diplopod millipedes, two pairs of spiracles occur in each segment (at bases of legs) and open into atria and numerous tracheae tubes penetrating organs or ending in the haemolymph (e.g. [3,50] and references therein). A pair of ventral abdominal cavities or chambers as observed in *E. potii* gen. et sp. nov. does not correspond to any of these morphologies.

Millipedes also possess paired circular gonopores (genital pores), opening on eversible or permanently everted devices, associated with the base of the second pair of walking legs (e.g. [52]). Vaccari *et al.* [20] recognized 'a circular structure slightly offset from mid-width of the second post-abdominal segment' as a possible gonopore. The absence of eversible or permanently everted devices in the fossils is not conclusive as the latter are rarely sclerotized (e.g. [52]), and as such have a weak preservation potential. However, the fact that euthycarcinoids lack appendages on their post-abdomen precludes the identification of the circular cavities seen in *E. potii* gen. et sp. nov. and *Apankura machu* (Vaccari *et al.*, 2004) as gonopores homologous to those of diplopods.

The other open abdominal ventral features known in arthropods are the chambers enclosing book lungs in arachnids (e.g. [53,54]) and those enclosing lamellate gills in eurypterids (most probably capable of subaerial breathing via a secondary respiratory structure that comprises the vascularized ventral wall gill-tracts (Kiemenplatten) of the branchial chamber [55]; note that the recent discovery of trabeculae on the dorsal surface of each gill lamella confirms that eurypterids were capable of subaerial breathing [56]). The cavities in *E. potii* gen. et sp. nov. are highly comparable in shape and position, particularly resembling those enclosing lamellate gills in eurypterids as figured by Laurie [47]. Together with the lack of appendage on these segments, a similar function as chambers enclosing respiratory organs (yet of unknown type and morphology) appears the most likely and

unique plausible interpretation for them. The enlarged branchial chambers of some terrestrial crabs, forming cutaneous brachial lungs from vascularization of the chamber wall (e.g. [4]), might be analogues to the cavities in *E. potii* gen. et sp. nov., as proposed for eurypterid Kiemenplatten [55].

The recognition of air-breathing adaptation in *E. potii* gen. et sp. nov., from a specimen preserved fully articulated (i.e. most likely with no or limited transport) in very shallow-water to subaerial sediments, provides additional evidence for an amphibious lifestyle for euthycarcinoids. A second independent evolution (as for the tracheal system in apulmonate arachnids, myriapods and terrestrial insects) of a system similar to the book lungs within Arachnida [54], or to the lamellate gills in eurypterids (Chelicerata), cannot be discarded. However, it is very unlikely that euthycarcinoids (stem myriapods, Mandibulata) developed the same organs, but rather different structures, supporting the possibility that the earliest terrestrial animals may have adopted unique respiratory strategies [1]. Our findings indicate that different respiratory strategies have been used during the marine-to-terrestrial transition in the myriapod lineage and suggest that the tracheal systems in myriapods replaced earlier respiratory organs.

Data accessibility. Data are available in the following Dryad Digital Repository at: https://doi.org/10.5061/dryad.cjsxksn2w [57].

Authors' contributions. J.D. rediscovered the fossil in collections and composed the geological background and figure 1. P.G. described the fossil and composed the rest of the paper with input from J.C.L., P.L.M. and L.B. P.G., R.A.W., P.L.M., V.M.E., U.B. and L.B. participated in the synchrotron analyses. P.G. and R.A.W. processed the synchrotron data. All authors interpreted and discussed the results.

Competing interests. The authors have no competing interests.

Funding. Portions of this research were funded by the France – Stanford Center for Interdisciplinary Studies Program and the Institute of Earth Sciences, University of Lausanne, which also supported open access publication costs.

Acknowledgements. Portions of this research were carried out at the Stanford Synchrotron Radiation Lightsource (CA, USA), a national user facility operated by Stanford University on behalf of the U.S. Department of Energy, Office of Basic Energy Sciences, and we thank the user support staff at SSRL. P.G. thanks G.D. Edgecombe for discussion. U.B. and L.B. acknowledge support from the France – Stanford Center for Interdisciplinary Studies Program, and P.G. from the Institute of Earth Sciences, University of Lausanne. We thank two anonymous referees for their reviews of the manuscript.

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
