## [Reviewer comments · Royal Society Open Science]

Review History

RSOS-201037.R0 (Original submission)

Review form: Reviewer 1

Is the manuscript scientifically sound in its present form?

Yes

Are the interpretations and conclusions justified by the results?

Yes

Is the language acceptable?

Yes

Do you have any ethical concerns with this paper?

No

Have you any concerns about statistical analyses in this paper?

No

Recommendation?

Accept with minor revision (please list in comments)

Comments to the Author(s)

Line 54: Di et al (reference 54) said that part of the scorpion book lung (the operculum) is of telopodal origin and contradicted homology with epipodites. This does not contradict a book lung / book gill homology unless one subscribes to book gills as epipodites (which has, admittedly, been endorsed by some morphologists, e.g., Suzuki and Bergström 2008).

Line 84: Springtails are said to respire by cuticular diffusion but this is the case only in some members of the group (poduromorphs and entomobryomorphs). Symphypleona have a pair of spiracles.

Lines 90-92: *Pneumodesmus newmani* has been redated as Early Devonian rather than Silurian (Suárez et al., 2017; Brookfield et al. 2020, *Historical Biology*).

Line 124: Johnson et al (reference 30) is cited in the context of terrestrial excursions but note that Shillito and Davies (2018) reinterpreted these trackways as subaqueous rather subaerial.

Lines 126-127 and 286-287: The “tube-like structures” in euthycarcinoids are not precisely associated with the paired sternal pores. Rather than say the tube-like structures resemble the tracheal respiratory system in “myriapods” (implicitly Myriapoda as a whole) it would be better to nuance this. The tube-like structures of euthycarcinoids resemble the sternal apodemes of Progoneata (there is a great drawing of them in each of the three progoneate groups in Dohle (1965, “Über die Stellung der Diplopoden im System”, *Zool. Anz., Suppl.* 28), but they are not known in Chilopoda. In Symphyla they are attachment sites for extrinsic leg muscles and longitudinal muscles and do not have an association with the respiratory system (as the spiracles are only on the head). It is only within Dignatha that they are apodemes forming the “tracheal pouch” (in millipedes and hexamerocerate pauropods; see Dohle 1997, fig. 23.4, “trt”, in the Fortey and Thomas book).

Lines 214-223, 281-283 (several instances): avoid “somite” when referring to euthycarcinoid tergites, as they cover multiple segments (somites). Likewise, “segments” in lines 232-234. Use “tergites”.

Line 307: Change “myriapods” to “diplopods” or “progoneate myriapods”. Some myriapods have an unpaired posterior gonopore that does not have an appendicular association (Chilopoda).

Trivial edits:

Line 73: add “are” so passage reads “are not rigid and are highly vulnerable”.

Line 76: “spiders” rather than “spider”.

Line 241: “represents” rather than “represent”.

Line 288: “loss” rather than “losses”.

Line 294: Reference 44 is consistent with two origins of the tracheal system in Chilopoda but not reference 45.

Line 294: “Symphyllan” rather than “symphylan”, and delete “pseudocentipede”, as it is a misnomer and is rarely used. The term “garden centipede” sometimes gets used for the agricultural pests within Symphyla, but no need to perpetuate “centipede” references for this group.

Line 501: “bubbles” rather than “bubble”.

Review form: Reviewer 2

Is the manuscript scientifically sound in its present form?

Yes

Are the interpretations and conclusions justified by the results?

Yes

Is the language acceptable?

No

Do you have any ethical concerns with this paper?

No

Have you any concerns about statistical analyses in this paper?

No

Recommendation?

Major revision is needed (please make suggestions in comments)

Comments to the Author(s)

This manuscript describes structures interpreted as adaptations for air breathing in a poorly preserved, poorly understood group of arthropods – euthycarcinoids. The novel aspect of the study is that the structures are delineated by geochemical imaging, particularly mapping of trace elements (As and Mn are shown in Fig. 2). These are compared to air breathing structures in other groups of arthropods.

The method assumes that the distribution of trace elements is controlled by and reflects the original anatomical structures of the organism. The diagenetic processes and history are not discussed at all. The authors should provide evidence that the structures that they are attributing to air breathing are not the nonbiogenic artifacts of some diagenetic process.

This is not the journal for a systematic description of a new genus. The section on systematics should be expanded and published in a journal focused on systematics. Removing systematics from the manuscript would allow space for better explanation of the processes (and products) of preservation.

The fact that the paper glues together -- rather than fully integrates -- euthycarcinoid systematics, arthropod anatomy, taxonomy, and history, and cutting-edge geochemical methods suggests that the authors have not identified a target audience on which to focus. The extensive and detailed discussion of air-breathing adaptations in different groups of arthropods is sufficiently crammed with anatomical and taxonomic jargon that its meaning and importance would elude all but arthropod specialists. Similarly, the geochemical methods and supplemental data would be opaque to non-geochemists.

The paper needs a thorough rewriting and reorganization to explain and highlight its contributions and significance to scientists with diverse perspectives.

If the following questions could be answered clearly, the paper would be excellent.

-- How and when in the taphonomic/diagenetic history were the trace elements incorporated?

- How does the diagenetic history relate to the (barely discussed) paleoenvironmental history? Significance of limonite?, sulfate?
- What do the structures suggest about air-breathing by eucyrcarionids in the Devonian? (simple diagram would help)
- How does this discovery about the group fit in with the environmental distributions (marine, fluvial, lacustrine, semi-aquatic, terrestrial) of euthycarcionids later in their history, such as recorded in the Permian of Antarctica?
- How do these adaptations – explained without jargon or illustrated with simple diagrams – relate to air breathing adaptations in other groups of arthropods, and what does this imply for the history of arthropods?

Decision letter (RSOS-201037.R0)

Dear Dr Gueriau,

The editors assigned to your paper ("A new Devonian euthycarcionid reveals the use of different respiratory strategies during the marine-to-terrestrial transition in the myriapod lineage") have now received comments from reviewers. We would like you to revise your paper in accordance with the referee and Associate Editor suggestions which can be found below (not including confidential reports to the Editor). Please note this decision does not guarantee eventual acceptance.

Note that in this case the expectation of myself as Subject Editor and the Associate Editor is that if you consider and address the referee comments in a thorough and reasonable way then it will be possible to accept your paper without needing further opinion from the referees. Please pay particular attention to the comments of one referee on what level of detail is appropriate to include in a RSOS paper and whether the paper will be accessible to readers outside of a particular specialist area.

Please submit a copy of your revised paper before 30-Jul-2020. Please note that the revision deadline will expire at 00.00am on this date. If we do not hear from you within this time then it will be assumed that the paper has been withdrawn. In exceptional circumstances, extensions may be possible if agreed with the Editorial Office in advance. We do not allow multiple rounds of revision so we urge you to make every effort to fully address all of the comments at this stage. If deemed necessary by the Editors, your manuscript will be sent back to one or more of the original reviewers for assessment. If the original reviewers are not available, we may invite new reviewers.

- Data accessibility

If you wish to submit your supporting data or code to Dryad (<http://datadryad.org/>), or modify your current submission to dryad, please use the following link:
<http://datadryad.org/submit?journalID=RSOS&manu=RSOS-201037>

- Competing interests

- Authors' contributions

- Acknowledgements

- Funding statement

Kind regards,
The RSOS Editorial Office
Royal Society Open Science
openscience@royalsociety.org

on behalf of Professor Rachel Wood (Associate Editor) and Peter Haynes (Subject Editor)
openscience@royalsociety.org

Reviewers' Comments to Author:

Reviewer: 1
Comments to the Author(s)

Line 54: Di et al (reference 54) said that part of the scorpion book lung (the operculum) is of telopodal origin and contradicted homology with epipodites. This does not contradict a book lung / book gill homology unless one subscribes to book gills as epipodites (which has, admittedly, been endorsed by some morphologists, e.g., Suzuki and Bergström 2008).

Line 84: Springtails are said to respire by cuticular diffusion but this is the case only in some members of the group (poduromorphs and entomobryomorphs). Symphypleona have a pair of spiracles.

Lines 90-92: *Pneumodesmus newmani* has been redated as Early Devonian rather than Silurian (Suárez et al., 2017; Brookfield et al. 2020, *Historical Biology*).

Line 124: Johnson et al (reference 30) is cited in the context of terrestrial excursions but note that Shillito and Davies (2018) reinterpreted these trackways as subaqueous rather subaerial.

Lines 126-127 and 286-287: The "tube-like structures" in euthycarcinoids are not precisely associated with the paired sternal pores. Rather than say the tube-like structures resemble the tracheal respiratory system in "myriapods" (implicitly Myriapoda as a whole) it would be better to nuance this. The tube-like structures of euthycarcinoids resemble the sternal apodemes of Progoneata (there is a great drawing of them in each of the three progoneate groups in Dohle (1965, "Über die Stellung der Diplopoden im System", *Zool. Anz., Suppl.* 28), but they are not known in Chilopoda. In Symphyla they are attachment sites for extrinsic leg muscles and longitudinal muscles and do not have an association with the respiratory system (as the spiracles are only on the head). It is only within Dignatha that they are apodemes forming the "tracheal pouch" (in millipedes and hexamerocerate pauropods; see Dohle 1997, fig. 23.4, "trt", in the Fortey and Thomas book).

Lines 214-223, 281-283 (several instances): avoid "somite" when referring to euthycarcinoid tergites, as they cover multiple segments (somites). Likewise, "segments" in lines 232-234. Use "tergites".

Line 307: Change "myriapods" to "diplopods" or "progoneate myriapods". Some myriapods have an unpaired posterior gonopore that does not have an appendicular association (Chilopoda).

Trivial edits:

Line 73: add "are" so passage reads "are not rigid and are highly vulnerable".

Line 76: “spiders” rather than “spider”.

Line 241: “represents” rather than “represent”.

Line 288: “loss” rather than “losses”.

Line 294: Reference 44 is consistent with two origins of the tracheal system in Chilopoda but not reference 45.

Line 294: “Symphylan” rather than “symphylan”, and delete “pseudocentipede”, as it is a misnomer and is rarely used. The term “garden centipede” sometimes gets used for the agricultural pests within Symphyla, but no need to perpetuate “centipede” references for this group.

Line 501: “bubbles” rather than “bubble”.

Reviewer: 2

Comments to the Author(s)

This manuscript describes structures interpreted as adaptations for air breathing in a poorly preserved, poorly understood group of arthropods – euthycarcinoids. The novel aspect of the study is that the structures are delineated by geochemical imaging, particularly mapping of trace elements (As and Mn are shown in Fig. 2). These are compared to air breathing structures in other groups of arthropods.

The method assumes that the distribution of trace elements is controlled by and reflects the original anatomical structures of the organism. The diagenetic processes and history are not discussed at all. The authors should provide evidence that the structures that they are attributing to air breathing are not the nonbiogenic artifacts of some diagenetic process.

This is not the journal for a systematic description of a new genus. The section on systematics should be expanded and published in a journal focused on systematics. Removing systematics from the manuscript would allow space for better explanation of the processes (and products) of preservation.

The fact that the paper glues together -- rather than fully integrates -- euthycarcinoid systematics, arthropod anatomy, taxonomy, and history, and cutting-edge geochemical methods suggests that the authors have not identified a target audience on which to focus. The extensive and detailed discussion of air-breathing adaptations in different groups of arthropods is sufficiently crammed with anatomical and taxonomic jargon that its meaning and importance would elude all but arthropod specialists. Similarly, the geochemical methods and supplemental data would be opaque to non-geochemists.

The paper needs a thorough rewriting and reorganization to explain and highlight its contributions and significance to scientists with diverse perspectives.

If the following questions could be answered clearly, the paper would be excellent.

- How and when in the taphonomic/diagenetic history were the trace elements incorporated?
- How does the diagenetic history relate to the (barely discussed) paleoenvironmental history? Significance of limonite?, sulfate?
- What do the structures suggest about air-breathing by eucarcinoids in the Devonian? (simple diagram would help)
- How does this discovery about the group fit in with the environmental distributions (marine, fluvial, lacustrine, semi-aquatic, terrestrial) of euthycarcinoids later in their history, such as recorded in the Permian of Antarctica?

-- How do these adaptations – explained without jargon or illustrated with simple diagrams – relate to air breathing adaptations in other groups of arthropods, and what does this imply for the history of arthropods?

Author's Response to Decision Letter for (RSOS-201037.R0)

See Appendix A.

Decision letter (RSOS-201037.R1)

Dear Dr Gueriau,

It is a pleasure to accept your manuscript entitled "A new Devonian euthycarcinoid reveals the use of different respiratory strategies during the marine-to-terrestrial transition in the myriapod lineage" in its current form for publication in Royal Society Open Science.

on behalf of Professor Rachel Wood (Associate Editor) and Peter Haynes (Subject Editor)
openscience@royalsociety.org

Appendix A

UNIL | Université de Lausanne
Institut des sciences de la Terre
bâtiment Géopolis bureau GEO-3430
CH-1015 Lausanne

Pierre GUERIAU
Institute of Earth Sciences
University of Lausanne
CH-1015 Lausanne, Switzerland

Lausanne, 22 August 2020

Ref.: Revision of manuscript RSOS-201037.

Title: *A new Devonian euthycarcinoid reveals the use of different respiratory strategies during the marine-to-terrestrial transition in the myriapod lineage*

Authors: Pierre Gueriau (*corresponding author*), James C. Lamsdell, Roy A. Wogelius, Phillip L. Manning, Victoria M. Egerton, Uwe Bergmann, Loïc Bertrand, and Julien Denayer

The Editors,

We hereby submit a revised version of our manuscript describing a new arthropod from the Devonian of Belgium, providing new insights into for the marine-to-terrestrial transition in the myriapod lineage.

We have carefully considered all reviewers' comments and have modified our manuscript accordingly (see our point-by-point responses to the reviewers' comments below). As requested, we paid particular attention to Reviewer 2's comments on "what level of detail is appropriate to include in a *RSOS* paper and whether the paper will be accessible to readers outside of a particular specialist area". Note that, however, while Reviewer 2 called for a simplification of the level of details, Reviewer 1 asked for highly detailed terminology corrections. We rather concur with Reviewer 1 and advocate that multidisciplinary studies such as the one performed in our paper require being scientifically correct in all involved fields (using proper scientific terminology). Nonetheless, we agree with Reviewer 2 that some reorganization would make our work clearer, which we have done to some extent. He/She also requested more details about the preservation and palaeoenvironment associated with the fossil, which we have provided. Finally, Reviewer 2 also commented on two other points that we find very questionable and have therefore decided not to address: (i) that *RSOS* "is not the journal for a systematic description of a new genus", which we do not think is true and prefer to leave up to you; (ii) He/She requested comparison with later euthycarcinoids and particularly a fossil from the Permian of Antarctica, for which age is actually poorly constrained and that some of us are convinced is not even an arthropod but a vertebrate.

We thank you for considering this revised version, and both reviewers for their comments that helped improving and clarifying our manuscript.

Yours sincerely,

Pierre GUERIAU
Phone: + 33 (0)1 69 35 81 06
Email: pierre.gueriau@unil.ch

Faculté des géosciences et de l'environnement
Institut des sciences de la Terre

Point-by-point responses to the Reviewers' comments and suggestions.

We present the *reviewers' comments in bold italics* and our responses in normal typeface.

Reviewer 1

Line 54: Di et al (reference 54) said that part of the scorpion book lung (the operculum) is of telopodal origin and contradicted homology with epipodites. This does not contradict a book lung / book gill homology unless one subscribes to book gills as epipodites (which has, admittedly, been endorsed by some morphologists, e.g., Suzuki and Bergström 2008).

Instead of “Line 54: Di et al (reference 54)”, the reviewer actually refers to Line 99: Di et al (reference 9)”. We clarified this point by modifying the sentence as follows:

“[...] suggesting that arachnid book lungs are derived from internalized book gills [10]. Gene expression data and embryology of modern scorpions however showed that part of the book lung (the operculum) is derived from the walking legs of abdominal appendages (telopodites), rather than from epipods (the outermost ramus of appendages) as for book gills [11].”

Line 84: Springtails are said to respire by cuticular diffusion but this is the case only in some members of the group (poduromorphs and entomobryomorphs). Symphypleona have a pair of spiracles.

We clarified this point by modifying the sentence as follows:

“[...] except for some springtails [...]”.

Lines 90-92: Pneumodesmus newmani has been redated as Early Devonian rather than Silurian (Suárez et al., 2017; Brookfield et al. 2020, Historical Biology).

The reviewer is absolutely right. This mistake on our part has been corrected, and the two mentioned references cited and added to the references list.

Line 124: Johnson et al (reference 30) is cited in the context of terrestrial excursions but note that Shillito and Davies (2018) reinterpreted these trackways as subaqueous rather subaerial.

The reviewer is absolutely right. In order to keep the message concise, we have chosen to drop the reference to Johnson et al.

Lines 126-127 and 286-287: The “tube-like structures” in euthycarcinoids are not precisely associated with the paired sternal pores. Rather than say the tube-like structures resemble the tracheal respiratory system in “myriapods” (implicitly Myriapoda as a whole) it would be better to nuance this. The tube-like structures of euthycarcinoids resemble the sternal apodemes of Progoneata (there is a great drawing of them in each of the three progoneate groups in Dohle (1965, “Über die Stellung der Diplopoden im System”, Zool. Anz., Suppl. 28), but they are not known in Chilopoda. In Symphyla they are attachments sites for extrinsic leg muscles and longitudinal muscles and do not have an association with the respiratory system (as the spiracles are only on the head). It is only within Dignatha that they are apodemes forming the “tracheal pouch” (in millipedes and hexamerocerate pauropods; see Dohle 1997, fig. 23.4, “trt”, in the Fortey and Thomas book).

We thank the reviewer for these precisions. We have nuanced the sentences as follows:

Introduction: “Several euthycarcinoids display a pair of ventral pores per pre-abdominal somite, located anteriorly to internal tube-like structures [25,32,33] resembling the sternal apodemes of progoneate myriapods.”

Discussion: “No pore or opening structure is visible in *E. potii* gen. et sp. nov., whereas some other euthycarcinoids exhibit ventral pores located anteriorly to internal tube-like structures [25,32,33] resembling the sternal apodemes that form the tracheal respiratory system within progoneate dignathan myriapods (these pores have alternatively been interpreted as vesicles minimizing water losses [25]).”

Lines 214-223, 281-283 (several instances): avoid “somite” when referring to euthycarcinoid tergites, as they cover multiple segments (somites). Likewise, “segments” in lines 232-234. Use “tergites”.

We used “somite” throughout our text because the specimen shows a ventral, internal view of the tergites for the cephalon and pre-abdomen, but an external view of the sternites (covering the rest of the somites) for the post-abdomen and telson. Following the reviewer’s comment, we made this clear in §3c, and accordingly replaced “somite” and “segment” throughout the text with “tergites” when referring to tergites.

Line 307: Change “myriapods” to “diplopods” or “progoneate myriapods”. Some myriapods have an unpaired posterior gonopore that does not have an appendicular association (Chilopoda).

Corrected

Trivial edits:

Line 73: add “are” so passage reads “are not rigid and are highly vulnerable”.

Corrected

Line 76: “spiders” rather than “spider”.

Corrected

Line 241: “represents” rather than “represent”.

Corrected

Line 288: “loss” rather than “losses”.

Corrected

Line 294: Reference 44 is consistent with two origins of the tracheal system in Chilopoda but not reference 45.

Corrected

Line 294: “Symphylan” rather than “symphilan”, and delete “pseudocentipede”, as it is a misnomer and is rarely used. The term “garden centipede” sometimes gets used for the agricultural pests within Symphyla, but no need to perpetuate “centipede” references for this group.

Corrected

Line 501: “bubbles” rather than “bubble”.

Corrected

Reviewer 2

This manuscript describes structures interpreted as adaptations for air breathing in a poorly preserved, poorly understood group of arthropods – euthycarcinoids. The novel aspect of the study is that the structures are delineated by geochemical imaging, particularly mapping of trace elements (As and Mn are shown in Fig. 2). These are compared to air breathing structures in other groups of arthropods.

The method assumes that the distribution of trace elements is controlled by and reflects the original anatomical structures of the organism. The diagenetic processes and history are not discussed at all. The authors should provide evidence that the structures that they are attributing to air breathing are not the nonbiogenic artifacts of some diagenetic process.

The issue raised by the reviewer that the “structures attributed to air breathing” indeed reflect ‘true’ anatomical features and not the result of diagenetic processes is indeed central here, and we apologize for not having been clear on this point.

No one will contest the fact that cuticular remains of a largely articulated euthycarcinoid are preserved on the slab investigated herein. The fact that elemental distributions strictly match these cuticular remains (or their outline) clearly indicates that the distribution of trace elements has been controlled by and reflects original anatomical structures of the organism. But *admittedly* not the entire anatomy, as soft-tissues and most appendages are not preserved; only cuticular remains. In contrast, elemental distributions that do NOT match the fossil anatomy are clearly associated with the sedimentary matrix where they do not exhibit organized morphologies and seem randomly distributed.

Regarding the pair of 2mm-large spherical air-breathing cavities revealed by elemental mapping, it is therefore very unlikely that diagenetic processes would produce, only there, such spherical and symmetrical morphologies.

In order to make this point clearer in our manuscript, we have included these arguments within a new “Preservation and palaeoenvironment” section (§3.3), see below.

This is not the journal for a systematic description of a new genus. The section on systematics should be expanded and published in a journal focused on systematics. Removing systematics from the manuscript would allow space for better explanation of the processes (and products) of preservation.

Several much longer papers published in *RSOS* have described new genus, and to our knowledge there is no space issue here. Therefore, unless requested by the Editors, we prefer to keep the systematic description of the fossil in our manuscript.

We have also expanded our section relating to fossilization processes, see below.

The fact that the paper glues together -- rather than fully integrates -- euthycarcinoid systematics, arthropod anatomy, taxonomy, and history, and cutting-edge geochemical methods suggests that the authors have not identified a target audience on which to focus. The extensive and detailed discussion of air-breathing adaptations in different groups of arthropods is sufficiently crammed with anatomical and taxonomic jargon that its meaning and importance would elude all but arthropod specialists. Similarly, the geochemical methods and supplemental data would be opaque to non-geochemists.

We recognize that our work is highly multidisciplinary, which requires being scientifically correct in different fields (using proper scientific terminology). Just as “geochemical methods would be opaque to non-geochemists”, oversimplifying the geochemical methods and data will frustrate geochemists. Likewise, simplifying anatomical and taxonomic terminology will frustrate palaeontologists and systematicists, as illustrated by several comments from reviewer #1.

Regarding the “target audience on which to focus”, it appears clearly in our title and introduction, and it is the only subject of our discussion: air-breathing in euthycarcinoids. Technical details about the geochemical analyses performed appear only in the Material and Methods section, and very minimally in the description of the results.

The paper needs a thorough rewriting and reorganization to explain and highlight its contributions and significance to scientists with diverse perspectives.

Following our response to the reviewer's previous comment, we think that a thorough rewriting as proposed by the reviewer would be at the expense of scientific precision and of important methodological details that support the conclusions.

Nonetheless, we agree with the reviewer that some reorganization would make our work clearer on several places:

- We have changed the numbering of the section to add a proper "Results" sections that, we believe, helps clarifying the focus of our work.
- In this "Results" section, we now gather all information on preservation and palaeoenvironment into a proper subsection, responding to part of the reviewer's comment below.
- The section "Additional anatomical details revealed by trace metal distributions" now only contains anatomical details, and no chemistry.

If the following questions could be answered clearly, the paper would be excellent.

(1) How and when in the taphonomic/diagenetic history were the trace elements incorporated?

(2) How does the diagenetic history relate to the (barely discussed) palaeoenvironmental history? Significance of limonite?, sulfate?

(3) What do the structures suggest about air-breathing by eucyrcarcinoids in the Devonian? (simple diagram would help)

(4) How does this discovery about the group fit in with the environmental distributions (marine, fluvial, lacustrine, semi-aquatic, terrestrial) of euthycarcinoids later in their history, such as recorded in the Permian of Antarctica?

(5) How do these adaptations – explained without jargon or illustrated with simple diagrams – relate to air breathing adaptations in other groups of arthropods, and what does this imply for the history of arthropods?

We thank the reviewer for this very positive comment.

Questions (1) and (2) are now fully addressed in our reorganized "preservation and palaeoenvironment" section (§3.3).

Questions (3) and (5) are the object of the discussion, and the conclusion of our work. Providing diagrams or reconstructions would be too speculative.

Regarding question (4), any further discussion would also be only speculative as later euthycarcinoids do not preserved remains of respiratory organs, even the well-preserved fossils from the Triassic 'grès à Voltzia'. As for the freshwater fossil from the Permian of Antarctica, the authors of the paper describing this fossil explain that "the age of the Pagoda Formation at Mt. Butters is poorly constrained; the lower part of the formation could be latest Carboniferous (Gzelian) to earliest Permian (Asselian)", so there is absolutely no guarantee that the fossil is really Permian in age. Moreover, at the risk of upsetting the reviewer, some of us (PG and JCL) are absolutely convinced that this fossil is not a euthycarcinoid but something completely different.